# Evaluating the Effectiveness of Visuospatial Memory Stimulation Using Virtual Reality in Head and Neck Cancer Patients—Pilot Study

**DOI:** 10.3390/cancers15061639

**Published:** 2023-03-07

**Authors:** Anna Serweta-Pawlik, Maciej Lachowicz, Alina Żurek, Bill Rosen, Grzegorz Żurek

**Affiliations:** 1Department of Physiotherapy, Wroclaw University of Health and Sport Sciences, J. I. Paderewskiego 35, 51-612 Wroclaw, Poland; 2Department of Biostructure, Wroclaw University of Health and Sport Sciences, J. I. Paderewskiego 35, 51-612 Wroclaw, Poland; 3Institute of Psychology, University of Wroclaw, Dawida 1, 50-527 Wroclaw, Poland; 4Neural Injury Center, University of Montana, 32 Campus Drive, Missoula, MT 59812, USA

**Keywords:** working memory, oncological patients, virtual reality, visuospatial memory, Cancer-related Cognitive Impairment

## Abstract

**Simple Summary:**

The scientific literature is increasingly drawing attention to the adverse effects of cancer treatment on cognitive function. Dedicated mental health support programs for patients diagnosed with cancer are using innovative technologies. The purpose of our study was to evaluate the effectiveness of using virtual reality to stimulate visuospatial memory. The study group included head and neck cancer patients who rarely participate in clinical trials. The results of the pilot study provided information on the feasibility of using virtual reality equipment in this group of patients and showed a favorable trend of changes in the working memory in individual patients.

**Abstract:**

The prevalence of Cancer-related Cognitive Impairment (CRCI) in cancer patients necessitates the search for methods to help stimulate cognitive function. An innovative and repeatedly used method in oncology departments is virtual reality (VR). To date, no one has used VR for head and neck cancer patients in an attempt to stimulate their working memory. The goal of our research is to intervene with off-the-shelf VR applications in HNC patients to lower the risk of CRCI. Twenty-two patients with head and neck cancer were enrolled in this pilot study to characterize their visuospatial memory capacity, a form of working memory. During the oncological treatment, the patient had 30 min sessions, three times a week, using publicly available applications of VR. No significant changes were observed in the pre- and post-study. The individual patient results present a favorable trend of changes in their working memory after the intervention: despite oncological treatment, visual-spatial memory did not deteriorate in 88% of patients, including 28% of patients with higher scores than before the intervention. In this pilot study, VR was safely used in patients with head and neck cancer with no negative side effects. The use of VR may prevent CRCI in most HNC patients and, in some, VR may improve their cognitive functioning. A more rigorous study with larger numbers and controls is advised.

## 1. Introduction

The global incidence of cancer is significant. Moreover, there is an increasing trend in the incidence of various types of cancer from year to year [1]. One of the most common malignancies is head and neck cancer (HNC). This category includes a variety of cancers, such as cancers of the lips, mouth, tongue, salivary glands, pharynx, larynx, and nasal cavity. These cancers account for approximately 900,000 new cases and more than 400,000 deaths annually worldwide [2]. Regardless of the location of the cancerous lesion, treatment requires a multidisciplinary approach. It most often involves surgery, chemotherapy (CHT), and radiation therapy (RT) [3]. The most common complications following the treatment of HNCs are the impairment of the tissues located around the cancerous lesion. The direct application of RT to brain and head and neck cancers can also result in a debilitating cognitive syndrome [4]. Further, there is a growing body of scientific literature addressing cognitive impairment as an acute and long-term consequence of CHT and RT. The colloquial term chemobrain, or Cancer-related Cognitive Impairment or CRCI, describe the cognitive dysfunction of patients who have undergone cancer treatment and experience a variety of higher level central nervous system issues, including fatigue, which hinders their return to normal functioning [5]. With the knowledge that that the processes of neurogenesis do not end right after birth, but continue through life, it is imperative to look for methods that strengthen the central nervous system for cancer patients. A promising solution to support cognitive function is virtual reality (VR). VR technology controls the progression of stimulus strength, allows for the ability to interact, includes feedback, and provides a safe environment for the patient [6]. Numerous studies indicate that VR is an effective tool for both diagnosing and improving working memory, attention, and other executive functions [7,8,9,10,11]. One of the most impaired cognitive abilities in patients undergoing cancer treatment is working memory. Working memory is necessary for a variety of complex cognitive tasks, such as learning, reasoning, and language comprehension [12]. A functioning working memory allows the individual to make informed decisions about treatment, adherence to recommendations, and improves their quality of life and, ultimately, the chances of patient survival [13], hence the importance of maximizing cognitive performance throughout cancer treatment. To the best of our knowledge, to date, there have been no studies using VR to stimulate cognitive function in HNC patients. The purpose of this pilot study was to use publicly available apps in HNC patients to determine whether VR can maintain and/or stimulate cognitive function during treatment of their cancer.

## 2. Material and Methods

The study was conducted at the Lower Silesian Oncology Center in Wroclaw in two periods: XI 2019–III 2020 and IV–VI 2022 (the break was due to COVID-19 pandemic restrictions). The inclusion criteria were a diagnosis of cancer in the head or neck area and a consent to participate in the study; the exclusion criterion was a diagnosed neurodegenerative brain disease (e.g., Alzheimer’s disease) or stroke. Twenty-eight patients diagnosed with cancer of the head or neck region were included in the study. The study was eventually completed by 18 patients: 11 men and 7 women, ME = 64, SD = 10 (Figure 1). The patients received conventional fractionation 5 times a week, once a day, and the size of the applied irradiation dose ranged between 50 and 70 Gy. Twelve patients additionally received chemotherapy. Details of the characteristics of the subjects included in the study can be found in Table 1. Patient memory testing took place on the day of admission to the hospital (pre-test) and on the day of discharge from the hospital (post-test). The CORSI Block-Tapping test consisted of nine cubes displayed on a monitor screen, placed in a random order. According to the Vienna Test System instructions for the S1 version of the test, the subject is required to reproduce the same sequence of cubes that was previously identified randomly by an automatic pointer on the monitor screen. The test begins with a sequence of three selected cubes, and the length of the series gradually increases. The maximum length of the series for which the test subject obtained two correct repetitions was accepted as the visuospatial working memory score (VSWM), which ranged between of 3–10. The measured parameters were the time required to complete the task, the number of correct sequences, the number of incorrect sequences, any omitted cubes, and any sequence errors. The VR stimulation took place 3 times a week throughout the patient’s hospital stay (6–7 weeks); the exposure time was 15–30 min, depending on the type and complexity of the application. The applications were sourced from the publicly available Oculus Rift movie database. The schedule of the apps used in the project is shown in Appendix A.

The study was approved by the Bioethics Committee at the Wroclaw University of Health and Sport Sciences No. KB-105/2017 and registered on the Australian New Zealand Clinical Trials Registry platform (ANZCTR, number ACTRN12619001279112).

All of the analyses, which included a Whitney’s U-Man test, Wilcoxon’s paired rank order test and Spearman’s correlation, were performed in Statistica (ver. 13.1) at the Biostructure Research Laboratory of Wroclaw University of Health and Sport Sciences (ISO 9001 certificate). The *p*-value was taken at the significance level of *p* < 0.05.

## 3. Results

The Mann-Whitney U test showed no significant differences between the sexes, age, and education of the subjects (Table 2).

The nonparametric Wilcoxon paired rank-order test (*p* < 0.05000) did not prove a change from the pre- to the post-intervention scores (Table 3). The patients’ individual scores, however, changed from the first study (Table 4). In 88% of the patients, the visual-spatial memory did not deteriorate after several weeks of cancer treatment. Despite the lack of statistical significance, it is noteworthy that in 28% of the subjects, the scores were higher after the intervention with VR. For 11% of the subjects, the score is lower in the second study, which is also not statistically significant (Table 3). Despite the lack of statistical significance (*p* = 0.09954), in the post-intervention testing, 67% of the patients improved their number of accurate responses (Table 4).

### Correlations

The Spearman’s correlation in the first study showed an almost full correlation (rs = 0.979) between the VSWM and accurate responses (Table 5). A higher number of accurate responses thus determined a higher level of VSWM. The direct block memory span was highly correlated (rs = 0.659) with the time it took to complete the task. The longer the test took, the higher the VSWM score achieved. The duration of the test correlated with the number of both right and wrong answers. A longer test time allowed for more correct answers, but also more incorrect answers.

In study two, the correlation of the VSWM and accurate answers was also almost complete (rs = 0.955). The numbers of incorrect and omitted responses over time also showed a significant correlation. The longer the task time, the more incorrect and omitted responses there were (rs = 0.688). There was also a moderate correlation between incorrect and omitted responses. A negative significant correlation (rs = −0.563) occurred between the duration of the task and the number of the incorrectly marked answer order (Table 5).

The correlations from both of the studies indicate that the VSWM depends primarily on the number of correctly answered questions.

## 4. Discussion

The 2021 HNC data estimates that half a million people worldwide won their battle with cancer [2]. Thus, along with this high survival rate, an important issue to consider is the survivor’s quality of life (QOL) during and after cancer treatment. QOL is dependent on the level of an individual’s cognitive functioning [14]. Unfortunately, cognitive functioning during cancer treatment is affected by a number of factors, which may include: age, toxicity issues related to the treatment strategies, psychiatric disorders, and the lack of rehabilitation interventions that may either ameliorate cognitive decline or enhance their capacity.

The risk factor of age contributes not only to the higher incidence of cancer, but also to other diseases that directly cause cognitive deterioration. Among cancer patients over the age of 65, between 3.8–7% have dementia [15]. Moreover, cognitive impairment arises in approximately 36% of cancer patients [16]. The examination of cognitive function in cancer patients is not common. A prospective study by Eriksen et al. (2022) indicates that the problem is common and underdiagnosed among the elderly [17]. Cognitive impairment can be easily overlooked, and the consequences of this oversight can be expressed in the patient’s decision-making capabilities, treatment adherence, self-care, other aspects of QOL, and ultimately, even survival [13]. According to a study by Raji et al. (2008), cognitive impairment at the beginning of cancer treatment was associated with a high mortality rate, and this was mainly due to non-cancer causes, with no relation to the stage at which the patient was diagnosed with cancer [15]. In our study, the inclusion criterion was not limited to patients over the age of 65 with pre-existing cognitive deficits for the sake of the possible later occurrence of nervous system disorders because a key determinant of cognitive health is also the patient’s mental health. The prevalence of depression among patients with head and neck cancer (HNC) ranges from 13–40% at the time of diagnosis, from 25% to 52% during treatment, and between 11–45% for the first six months after completing treatment [18]. Patients with HNC are particularly prone to developing relapse anxiety [19]. Anxiety and depression often result from symptoms specific to HNC, namely the possibility of the loss of speech, difficulty speaking, swallow dysfunction, and the distortion of a visible body part [18]. Unfortunately, not all effective anxiety-reducing interventions can be applied to HNC patients [20]. In hospital conditions, patients are often alienated, lack physical and cognitive exertion, experience a degradation in their social position, and all of this can result in a decline in cognitive and psychological functioning. Further, the importance of the relationship between cognitive and psychological health can be overlooked. It has been proven that a three week hospital stay can lower the patient’s IQ by as much as 20% [12]. For these reasons, at the design stage of the in-house study, it was considered unethical to create a control group that would be denied an intervention that could stimulate cognitive function at any stage of treatment, especially given the small sample size and its heterogeneity.

It is worth noting that cognitive impairment does not only affect people over 65 years of age. A factor that negatively affects mental health is the effect of CHT and RT, which is extensively described in the literature by the term CRCI. Various mechanisms cause RT to adversely affect patients’ cognitive function. Elevated levels of pro-inflammatory cytokines, endocrine dysfunction, extensive hypomyelination, and reduced neurogenesis appear to be the most significant [21]. The adverse effects of chemotherapy have also been repeatedly confirmed in scientific studies [22]. The most recent review by Vizer et al. (2022) includes studies about the cognitive impairment faced by people under 39 years of age with cancer located outside the central nervous system (CNS). The review found that CRCI affects approximately 26% of subjects of this age [23]. For all of the reasons cited above, patient age was not an inclusion criterion for our own study. In the present study, one patient under the age of 39 participated, but there were nine additional patients under the age of 65. Our results indicate that patient age did not have a statistically significant effect on the level of memory span, but due to the small sample, the results should be interpreted with caution. Nevertheless, among the two patients whose VSWM score decreased during their hospital stay, one was aged 64 and the other was 65.

Vizer et al. (2022), in their systematic review, highlight that of the 21 articles included in the review, only one was a study using an intervention, even though various types of activities can improve cognitive function. Interventions using digital methods, which can be a practical and effective option, were also not addressed [23]. The need for cognitive rehabilitation was noted by Dos Santos et al. [11], who demonstrated that training using computer-based methods led by a neuropsychologist is more effective than mental tasks performed at home or no mental exercises. Virtual reality has tremendous potential as a form of therapy, with proven effectiveness as a treatment modality for both mild cognitive impairment and dementia [24,25]. The advantages of this technology include that it is a relatively inexpensive form of technology, has a robust and controlled progression of difficulty, allows for opportunities for interaction and feedback, and can be accomplished in a safe environment [6]. The most cited disadvantages posed by the virtual environment are the required caution in its use in the elderly (risk of falling) and the possibility of motion sickness. Considering the available literature, the elderly is a large group of recipients of VR technology. VR rehabilitation techniques have been used in stroke, dementia, neurodegenerative diseases, fall prevention training, and as a diagnostic method of assessing cognitive dysfunction. In the rehabilitation of the cognitive function of oncology patients, VR is also effective, although clinical studies are still scarce and mainly involve breast cancer and investigations into mental health [26].

Zeng et al. (2022) used a specially designed virtual environment to stimulate the most impaired cognitive functions of patients with various cancers, excluding those in the CNS [27]. However, their investigation did not examine memory, despite the fact that it is one of the most vulnerable functions to cognitive impairment. What is noteworthy, however, is that the intervention was designed so that the patient received 30 min VR sessions ten times for the two weeks of their hospital stay. If the participant was only in the hospital for one week, then the sessions were held twice a day. Of the 17 cognitively impaired participants included, only nine completed the entire VR intervention. The reasons cited for not completing the study include high treatment burden, significant disease progression, severe side effects from chemotherapy, and early discharge from the hospital. It is worth considering whether such a high intensity of VR sessions is necessary to have an effect in improving cognitive function. In their meta-analysis, Zeng et al. (2019) showed that VR can be readily used in oncology as an interventional method for cognitive dysfunction, although there are no statistically significant differences, which may reflect the small numbers of study participants [28].

In the PubMed database, the only clinical study with VR in HNC involved fourteen cancer patients who utilized VR to potentially reduce the post-operative pain in the head and neck region [29]. The study also used a publicly available virtual game, Angry Birds. The intervention significantly reduced the subjective pain complaints and opioid use after surgery compared to a control group that used a smartphone game. Our pilot study was also designed to take advantage of commercially available apps for VR. We included as many as 16 apps in the study to provide patients with a variety of stimuli (Appendix A). The apps were chosen to either relax or, on the contrary, to activate and energize the patient to enliven the daily monotony [30].

VR requires the use of a head mounted display (HMD) to fully achieve the immersive effect. As VR users don a headset, this may appear to be a relative contraindication of the use of this technology in this patient population. However, Pandrangii et al. (2022) used VR in patients who had undergone extensive surgery in the head or neck region. Even with tissue damage in the head and neck region (either through surgery or RT) and limited mobility range of the neck, this was not a significant problem. Our study also found no consequential effects of the use of a HMD due to the underlying disease process. It appears that a limited range of head mobility is compensated for through eye movement and, further, in the present study, the use of a swivel chair allowed the patients to freely observe a wider range of images of the projected film.

Chronic post-operative pain in breast cancer survivors was studied by House et al. (2016). In this study, twice-weekly arm motor function training was performed in six depressed breast cancer survivors. Among the parameters studied, visual-spatial memory improved statistically significantly after eight weeks of training [31]. In our study, no statistically significant differences were observed between the pre- and post-tests. Based on the study by Martinez-Esparza et al. (2021), which used the CORSI test in two forms, one may wonder whether the backward subtype of the test would be more sensitive to the possible changes in the working memory functioning of cancer patients [32].

In pediatric patients with brain tumors, computerized cognitive training programs (CTTPs) that target specific cognitive domains are currently being used to improve cognitive function [33]. However, the research results are not homogeneous [34], although a positive trend of these interventions on neurocognitive deficits is generally emphasized. In our study, we focused on one cognitive domain—working memory, or more specifically, its subtype, visual-spatial memory—and the intervention (although sourced from a publicly available app database) was geared toward using—visual, auditory, and proprioceptive senses. Our study also did not show statistical significance, although there is a favorable trend in the post results. A review by Sciancalepore et al., 2022, suggested that CTTP improves cognitive functions, such as working memory, but some studies have revealed only transient positive effects, with a significant number of dropouts during ongoing follow-up (five of the nine studies included in this review were trials of less than 30 participants) [33]. In our case, we were unable to perform a longitudinal study to determine how visual-spatial memory changes long after completing treatment. This limitation was due to the loss of contact with the patients after hospital discharge (patients after completing hospital treatment are referred to the clinic closest to their residence for follow-up) and the pandemic situation. However, the turnout of participation in our study was high—only one person was not included in the study due to physical malaise already present on the day of hospital admission, and one person who refused to participate in the final test due to severe dizziness and nausea after CHT and RT. Eight patients did not perform the post-test due to—discontinuation of the study due to epidemiological recommendations during COVID-19.

The above literature reviews and clinical trials highlight the beneficial effects of virtual reality on various aspects of patient functioning. Undoubtedly, VR is a technology that can be implemented for patients hitherto cautiously included in clinical trials. The results of our study, although without proven significant effects on working memory, continue to encourage the use of publicly available VR applications as part of cognitive stimulation, especially on larger numbers of participants.

### Limitations

The study presented here has some limitations. Due to the limited availability of participants and the high risk of them dropping out of the study due to the severity of their cancer treatment, one limitation is the small number of participants. Future studies should include a larger group of participants to compare the results with our study. In addition, it would be advised in future studies to add an additional endpoint study to standardize the patient outcomes, which, because of CHT and RT, may be disrupted on the day of the study (e.g., interrupted study due to nausea). Unfortunately, the lack of a control group limits the accurate interpretation of the results. For this reason, our work is currently presented as a pilot study. In future studies, a control group should be included to account for the effects of the intervention.

## 5. Conclusions

The results of the present study indicate that the use of publicly available virtual reality applications has the potential to affect the visuospatial memory in cancer patients. The authors suggest that VR equipment can be used to maintain cognitive function in head and neck cancer patients. The effectiveness of this form of therapy to improve cognitive function should be tested on a larger group of subjects.

## Figures and Tables

**Figure 1 cancers-15-01639-f001:**
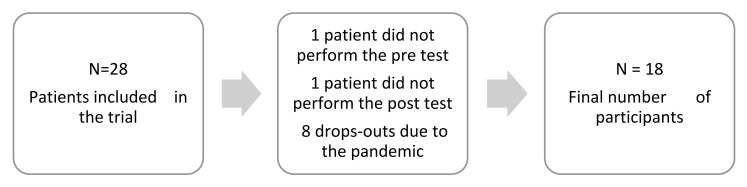
Flow chart. The process of study subject recruitment.

**Table 1 cancers-15-01639-t001:** Characteristics of patients included in the study.

	Sex	Age	Edu.	Diagnosis ICD-10 Code	TNM Classification	Treatment
P1	M	51	3	C01—Malignant neoplasm of base of tongue	T3 N0 M0	Induction radiotherapy 69.96 Gy/33 fr.
P2	M	38	3	C06.0 Cheek mucosa	T2 N3 M0	Radical radiotherapy 66.96 Gy/33 fr. combined with chemotherapy
P3	F	49	3	C09.9 –Tonsil, unspecified	T4a N2b M0	Radical radiotherapy 69.96 Gy/33 fr. combined with chemotherapy
P4	M	58	3	C15.0—Cervical part of esophagus	T3 N1 M0	Radical radiotherapy 30 Gy/10 fr.
P5	F	66	4	C.77—Secondary and unspecified malignant neoplasm of lymph nodes	T0 N3 M0	Radical radiotherapy 69.96 Gy/33 fr. combined with chemotherapy
P6	F	66	4	C32.8—Overlapping lesion of larynx	T4 N2 M0	Radical radiotherapy 66.96 Gy/33 fr. combined with chemotherapy
P7	F	64	3	C13—Malignant neoplasm of hypopharynx	T3 N0 M0	Radical radiotherapy 69.96 Gy/33 fr. combined with chemotherapy
P8	M	67	4	C77.9—Lymph node, unspecified	T1 N3 M0	Radical radiotherapy 69.96 Gy/33 fr. combined with chemotherapy
P9	M	76	4	C06.0—Cheek mucosa	T1 N1 M0	Radical radiotherapy 60 Gy/30 fr.
P10	F	76	5	C32—Glottis	T4 N2 M0	Radical radiotherapy 66.96 Gy/33 fr. combined with chemotherapy
P11	M	49	5	C01—Malignant neoplasm of base of tongue	T4a N2b M0	Radical radiotherapy 66 Gy/33 fr. combined with chemotherapy
P12	F	62	2	C32.9—Larynx, unspecified	T4a N0 M0	Radical radiotherapy 69.96 Gy/33 fr.
P13	F	70	3	C03.1—Malignant neoplasm of lower gum	T4 N1 M0	Radical radiotherapy 69 Gy/33 fr. combined with chemotherapy
P14	M	67	1	C09.9—Malignant neoplasm of tonsil, unspecified	T3N2 M0	Radical radiotherapy 69.96 Gy/33 fr. combined with chemotherapy
P15	M	63	3	C09.9 Malignant neoplasm of tonsil, unspecified	T3N2 M0	Radical radiotherapy 69.96 Gy/33 fr. combined with chemotherapy
P16	M	65	3	C06.0 Cheek mucosa	T3 N2a M0(x)	Radical radiotherapy 66.96 Gy/33 fr. combined with chemotherapy
P17	M	64	3	C05—Malignant neoplasm of palate	T4 N2b M0	Radical radiotherapy 50 Gy/20 fr.
P18	M	63	3	C02—Malignant neoplasm of other and unspecified parts of tongue	T1 N3 M0	Radical radiotherapy 50 Gy/20 fr.

Abbreviations: Edu.—education level; 1, below primary; 2, primary; 3, vocational; 4, secondary; 5, post-secondary.

**Table 2 cancers-15-01639-t002:** Differences in subjects by age, gender, and education.

Study	Variables	*p*-Value		
		Age	Sex	Education
Pre	time (s)	0.53	0.53	0.74
	VSWM	0.82	0.82	0.53
	correct	0.86	0.86	0.67
	incorrect	0.25	0.25	0.48
	omitted	0.45	0.45	0.22
	sequence error	0.44	0.44	0.66
Post	time (s)	0.45	0.45	0.93
	VSWM	0.85	0.85	0.19
	correct	0.93	0.93	0.26
	incorrect	0.89	0.89	0.15
	omitted	0.46	0.46	0.44
	sequence error	0.96	0.96	1.00

**Table 3 cancers-15-01639-t003:** Comparison of results of pre- and post-study.

Variables	N	T	Z	*p*	Study	Median	Min	Max
time	18	49	1.59	0.11	pre	196	73	320
					post	228	80	307
VSWM	7	10.5	0.59	0.55	pre	4	2	5
					post	4	2	5
correct	15	31	1.65	0.09	pre	6	0	8
					post	5	1	9
incorrect	14	47	0.35	0.72	pre	4	3	6
					post	4	3	6
omitted	3	0	1.60	0.1	pre	0	0	1
					post	0	0	3
sequence error	16	63.5	0.23	0.81	pre	2	0	4
					post	2	0	4

**Table 4 cancers-15-01639-t004:** Individual results in the pre- and post-test and changes between the tests.

Study	P	Time (s)	D	VSWM	D	Correct	D	Incorrect	D	Omitted	D	Sequence Error	D
Pre	P1	186	+	4	+	6	+	4	+	0	+/−	3	+
Post		213		5		9		3		0		1	
Pre	P2	215	+	5	+/−	8	+	3	−	0	+/−	1	−
Post		239		5		9		4		0		3	
Pre	P3	205	+	5	+/−	8	−	4	−	0	+/−	2	−
Post		217		5		7		6		0		4	
Pre	P4	265	−	5	+/−	7	+	4	+	0	+/−	1	+/−
Post		245		5		8		3		0		1	
Pre	P5	246	−	4	+	6	+	6	+	0	+/−	0	−
Post		241		5		8		4		0		1	
Pre	P6	247	+	5	+/−	7	+	4	+	0	−	1	+
Post		307		5		8		3		1		0	
Pre	P7	219	−	4	+/−	6	+/−	5	+/−	0	+/−	3	+
Post		208		4		6		5		0		2	
Pre	P8	73	+	3	+	2	+	3	+/−	0	+/−	3	+
Post		131		4		5		3		0		2	
Pre	P9	320	−	4	+/−	6	+	6	+	0	+/−	3	+
Post		289		4		7		4		0		1	
Pre	P10	160	+	3	+	4	+	4	+/−	0	+/−	1	−
Post		190		4		5		4		0		3	
Pre	P11	137	−	4	+/−	5	+/−	4	+	0	+/−	4	+
Post		118		4		5		3		0		3	
Pre	P12	126	+	2	+	0	+	3	−	0	+/−	0	−
Post		179		3		2		4		0		1	
Pre	P13	139	+	3	+/−	2	+	4	−	1	+/−	1	−
Post		249		3		3		5		1		2	
Pre	P14	185	+	2	+/−	1	+	5	−	1	−	0	−
Post		278		2		2		6		3		2	
Pre	P15	147	+	2	+/−	1	+	3	−	0	−	2	+
Post		289		2		2		5		2		0	
Pre	P16	111	−	2	+/−	1	+/−	4	+	0	+/−	2	+/−
Post		80		2		1		3		0		2	
Pre	P17	223	+	3	−	3	−	5	+/−	1	+/−	2	+
Post		256		2		1		5		1		0	
Pre	P18	223	−	4	−	6	−	4	+	0	+/−	3	+
Post		80		2		1		3		0		2	

The D denotes a difference between post- and pre-study. The plus sign (+) denotes an improvement in the second study. The minus sign (−) denotes a decline in the score relative to the pre-study. A plus and minus sign (+/−) indicates no change between the first and second study.

**Table 5 cancers-15-01639-t005:** Relationship between variables in pre- (heat map A) and post-study (heat map B).

A	VSWM	Correct	Incorrect	Omitted	Sequence Error	Time (s)
VSWM		0.000 *	0.642	0.113	0.530	0.003 *
correct	0.000 *		0.502	0.121	0.584	0.001 *
incorrect	0.642	0.502		0.188	0.955	0.012 *
omitted	0.113	0.121	0.188		0.234	0.821
squence error	0.530	0.584	0.955	0.234		0.793
time	0.003 *	0.001 *	0.012 *	0.821	0.793	
**B**	**VSWM**	**Correct**	**Incorrect**	**Omitted**	**Sequence error**	**Time (s)**
VSWM		0.000 *	0.324	0.062	0.649	0.628
correct	0.000 *		0.446	0.160	0.889	0.293
incorrect	0.324	0.446		0.044 *	0.716	0.002 *
omitted	0.062	0.160	0.044 *		0.078	0.002 *
sequence error	0.649	0.889	0.716	0.078		0.015 *
time	0.628	0.293	0.002 *	0.002 *	0.015 *	

* *p* < 0.05.

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
