# Peer review of "Evaluating the Effectiveness of Visuospatial Memory Stimulation Using Virtual Reality in Head and Neck Cancer Patients—Pilot Study"

_cancers, 2023, doi:10.3390/cancers15061639_

Round 1
Reviewer 1 Report
It is a very interesting and well written manuscript.
I suggest using and emphasizing in the title and text the aspect of the work as a pilot work because the 18 patients are not a representative group for the population of patients with head and neck cancer.
Author Response
Response to Reviewer 1 Comments
We thank the reviewer for highlighting these important points of our study. We believe this will greatly enhance the article. We have modified and hopefully clarified what requested.
Point 1: I suggest using and emphasizing the title and text the aspect of the work as a pilot work because the 18 patients are not a representative group for the population of patients with head and neck cancer.
Response: As suggested, we have indicated in the title of the article and in the text that the work presented here refers to a pilot study.

Reviewer 2 Report
see attached file for my specific comments

Author Response
Response to Reviewer 2 Comments
We thank the reviewer for highlighting these important points of our study. We believe this will greatly enhance the article. We have modified and hopefully clarified what requested.
Point 1: In the introduction it states that “One of the most common malignancies is head and neck cancer (HNC)” yet no reference is given. I am not sure that is accurate. The incidence is increasing but here in the United States, head, and neck cancer accounts for only 3 percent of malignancies, with approximately 66,000 cases annually and 15,000 deaths, so it is not one of the most common malignancies. I have provided a reference Gormley M, Schache A, Ingarfield K, et al. Reviewing the epidemiology of head and neck cancer: definitions, trends, and risk factors. Br Dent J. 2022; 233(9): 780–786.
Response: A footnote on the frequency of malignancies follows the statement of which cancers we have included in the head and neck cancer category: One of the most common malignancies is head and neck cancer (HNC). This category includes a variety of cancers, such as cancers of the lips, mouth, tongue, salivary glands, pharynx, larynx, and nasal cavity. These cancers account for about 900,000 new cases and more than 400,000 deaths annually worldwide [2 - Global Cancer Observatory]. According to Reviewer 2's attached review by Gormley M, Schache A, Ingarfield K, et al, the category of head and neck cancers is interpreted differently, so in the introduction we emphasized that in our study we adopted a classification that includes esophageal and thyroid cancers and the incidence of these cancers is determined by the cited incidence and death statistics.
Point 2: The description of education levels seems odd. Here are the described levels: 1, education below primary; 2, primary education; 3, vocational education; 4, secondary education; 5, obtaining a master’s or engineering degree. I think it would be better to just say 1- below primary; 2-primary; 3 vocational, 4 secondary, 5 post-secondary (rather than “obtaining a master’s or engineering degree”. The way this is stated is confusing. Are you equating an undergraduate degree in engineering to a masters or doctoral degree in other sciences? That is not a valid comparison.
Response: We agree with Reviewer 2 that the description of education is not sufficiently presented. Reviewer 2’s proposed description was included in the manuscript.
Point 3: Tables 3, 4, and 5 are very challenging to read. Could this data be converted to bar graphs or some other visual presentation. Once these are in a print journal, they will be very hard to read.
Response: After reviewing the tables again, as suggested by Reviewer 2, we decided to improve the tables to make them clearer. Table 5 has been replaced with a heat maps.
Point 4: The discussion does go over a lot of literature review. This is helpful but does add significantly to the length of the paper. If word count is a concern, then I would suggest streamlining the discussion section to focus more directly on the novel data contained in this study. If length of paper is not a concern, then I would leave in
Response: Due to the lack of studies in the literature that are similar to the work presented here, each time we cited a review or other literature item we related the cited study results to our own work, particularly the way the study was designed and the results obtained. Since we treat our study as a pilot study, we hope that the present work will serve as a good reference for other literature items and subsequent studies.
